# Characterization of Airfreight-Related Logistics Firms in the City of Cape Town, South Africa

**Masilonyane Mokhele** [1,*] and **Tholang Mokhele** [2]

[1] Department of Urban and Regional Planning, Faculty of Informatics and Design, Cape Peninsula University of Technology, Cape Town 8000, South Africa

[2] Geospatial Analytics, eResearch Knowledge Centre, Human Sciences Research Council, Pretoria 0001, South Africa; tamokhele@hsrc.ac.za

[*] Correspondence: mokhelem@cput.ac.za; Tel.: +27-21-440-2246

**Abstract:** *Background*: Airports are essential drivers of spatial development; hence the placement of logistics facilities relative to airports is a topical subject. Despite the wealth of the literature on the subject, relatively little is known about the airfreight catchment of airports. To contribute to the existing knowledge, the paper used the study area of the City of Cape Town municipality, South Africa, to address three research objectives, namely analysis of factors that influence the placement of logistics firms in the municipality, analysis of the linkages of the logistics firms with Cape Town International Airport (CTIA), and analysis of the association between airfreight-related firms and the general attributes of logistics firms in the municipality. *Methods*: The study hinged on a quantitative design, which included a survey and spatial analysis. A total of 110 logistics firms were sampled through a stratified random sampling technique, and 66 firms participated in the telephonic interviews conducted in October and November 2021. Survey data were analyzed using Stata, and spatial analysis was undertaken using ArcGIS 10.8 and QGIS 3.16. *Results*: It was discovered that a quarter of the respondent logistics firms utilized CTIA for airfreight purposes. At a municipal scale, the potential airfreight catchment of CTIA extended to about a 20 km radius of the airport. *Conclusions*: In formulating the spatial plans, the planning authorities are encouraged to take cognizance of the possible extent of the catchment, wherein airfreight-related firms do not necessarily locate near the airport.

**Keywords:** airfreight; Cape Town; Cape Town International Airport; logistics; airfreight catchment; South Africa





## 1. Introduction

Airports are crucial drivers of spatial development [1], particularly in the contemporary age characterized by deepening globalization, e-commerce, and logistics processes [2,3]. Therefore, the placement of logistics facilities relative to airports is a topical subject that transcends several fields of study. The literature on this subject can be categorized into three main focus areas: the concentration of logistics facilities in the vicinity of airports, the role of air transport and airports in metropolitan areas and regions becoming logistics hubs, and the airfreight catchment of airports [4]. This literature could also be linked to the normative spatial planning models of airport-led development [5], epitomized by the aerotropolis, which conceptualizes the airport and the surrounding airport city as the centre of development in a metropolis or region [6]. The models, which are interpreted and applied differently in various parts of the world, have become buzzwords used in spatial planning policies [7].

Although there is a wealth of the literature on airports and the positioning of logistics facilities, relatively little is known about the airfreight catchment of airports. Most research focuses on analyzing logistics facilities near airports [8]. The findings of such studies are usually used in the marketing strategies devised by airport authorities, government

agencies, and other stakeholders to promote urban development in the environs of airports. In the manner of the models of airport-led development, policymakers and spatial planners typically assume that economic activities understood to be airport-related (for instance, logistics firms) tend to be situated geographically proximate to airports. Showing that this understanding is not always accurate, numerous airports have failed to transform into airport cities [9], aerotropolis [1], or idealized urban forms informed by other derivatives of the models of airport-led development. Relatedly, there are instances where logistics parks established near airports are less successful in attracting logistics facilities compared to parks positioned elsewhere [10]. This discrepancy between the normative aspirations of the economic activity composition of the airport environs and the actual development patterns calls for an extension of the empirical analysis to include logistics facilities situated beyond the immediate surroundings of airports.

Focusing on the study area of the City of Cape Town municipality in South Africa (refer to Section 3.1), the paper aims to characterize airfreight-related logistics firms relative to non-airfreight-related firms towards determining the airfreight catchment of Cape Town International Airport at a metropolitan scale. This research aim is achieved by addressing the following objectives:

- Analysis of factors that influence the placement of logistics firms in the City of Cape Town;
- Analysis of the relationship between logistics firms and Cape Town International Airport; and
- Analysis of the association between airfreight-related logistics firms and the general attributes of logistics firms in the City of Cape Town.

The rest of the paper is structured as follows: Section 2 provides an overview of the literature on the relationship between airports and the surrounding areas, the airfreight catchment of airports, and factors that influence the location-choice decisions of logistics firms. Section 3 focuses on the data collection and analysis methods used to address the research objectives. Section 4 discusses the results of the analyses conducted. Section 5 concludes the paper.

## 2. Literature Review

Historically, discussions about air transport largely revolved around passenger concerns instead of airfreight or air cargo matters, partly because airfreight was considered a by-product of the air passenger service [3,11–13]. It is acknowledged that airlines and freight forwarders still utilize the belly of passenger aircraft to transport cargo [14,15], hence passenger air transport is critical in the logistics industry [14]. In light of the bias towards air passenger-related matters, there is a vast literature on the catchment of airports pertaining to the place of origin of passengers, including the underlying air passenger travel behavior [16–18]. There is, however, a relative paucity of research on the airfreight catchment of airports. To identify aspects that require improvement in the existing knowledge on the airfreight catchment of airports, the literature presented in this section focuses on three interrelated aspects. These are the relationships between airports and the surrounding areas and broader territories, the determination of the airfreight catchment of airports, and factors that influence the location-choice decisions of logistics firms.

### 2.1. Relationship between the Airport and the Surrounding Areas

A common theme in the airport literature is the agglomeration of airport-related economic activities around airports [19], where, with the rise of air transport, airports are regarded as functional anchors in cities and regions [20]. In this regard, a region can be categorized into several zones in the manner of the normative models of airport-led development, particularly the aerotropolis (Figure 1).

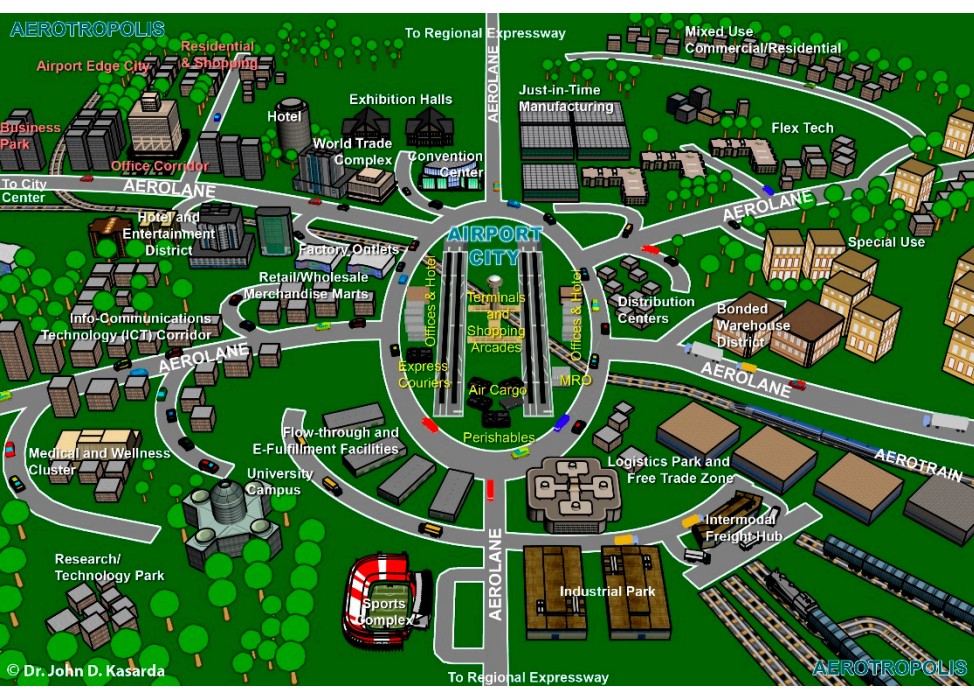

**Figure 1.** Aerotropolis schematic [21].

The first zone of the region (referred to as the airport city) includes the airport area and the associated businesses situated on the airport landholding. On-airport employment provides supporting functions such as airline management and aircraft maintenance [20]. The second zone encompasses the concentration of aviation-linked establishments situated beyond the airport land. This zone may attract aviation-oriented manufacturing firms that require air transport for time-sensitive goods with a high value-to-weight ratio [20]. The third zone includes a wide range of businesses [22], which may or may not be related to the airport's aviation services.

With a specific focus on freight transport and logistics, Boloukian [14] argues that logistics-related development forms a radial network around airports. The airport, the centroid of this network, propagates logistics establishments and stimulates logistics companies to position some components of their business geographically close to the airport. The resultant three main categories of logistics activities and their linkages can be summarized as follows: logistics activities that are directly linked to the airport's cargo services and operations; supplementary logistics activities that have close access to the airport services essential to the business and supply chain processes; and unconnected logistics activities, which benefit from the economic advantages of being positioned in the proximity of industrial zones and businesses around the airport [14].

According to Boloukian [14], the role of aviation in freight transport and logistics networks provides the platform for economic development within the airport catchment. Since the airport is potentially the radial point of development, logistics facilities could interact directly and indirectly with the airport. Some research on the growth of economic activities around airports has found that, in some instances, businesses that require or benefit from air transport seek locations that are near the airport, within traveling distances of up to 30 min [23,24]. The development of airport industrial parks has also resulted from the interactions between the airport and the surrounding areas. This is particularly relevant in the realm of logistics, where the reliability of supply chains is decisive and meaningful to the operations of logistics establishments [14]. On the landside, although last-mile delivery companies and trucking businesses do not necessarily have to be positioned near the airport, their presence there is essential for the efficiency of the cargo market [3].

## 2.2. Determination of the Airfreight Catchment of Airports

Diverse approaches are utilized to ascertain the catchment area of airports. Empirical studies employ a combination of pre-defined criteria, which include investigating the impact of a subject airport or, in the context of multi-airport systems, the analysis of the competition between airports. Each interpretation of a catchment area is based on different analyses of the airport's possible influence [25].

Traditionally, the airport catchment was measured by establishing radii of geographical distance around the airport or by calculating the travel time from given points to the airport. Although the approaches based on radii may be simple to interpret and apply, they have critical limitations because they do not consider other factors that could be useful in ascertaining the airport's sphere of influence. The following shortcomings are noted: firstly, the outcome of the analysis based on radii presents a static picture of the airport catchment area in which changes in the factors that influence the utilization of a particular airport do not affect the extent or scope of the catchment area. Secondly, the market share within the catchment area remains unclear, ignoring that the market share may decrease with the increasing distance from the airport. Thirdly, the catchment area based on simple radii is unrealistically assumed to be the same for every destination [26]. Because of these shortcomings, other researchers utilize alternative approaches to measure the catchment area of airports. For instance, Alves et al. [25] focused on a broader set of indicators, such as the ability of an airport to attract cargo and passengers, the quality of the airport services offered, and the impact of the airport's activities on the surrounding areas.

Against the backdrop of airports performing a crucial function for their immediate surroundings and larger territories [27], the literature on the airport catchment spans multiple geographical scales, namely airports and their immediate surroundings, metropolitan areas, functional regions, countrywide locations, areas across national borders, and even across continents. At a local level, and as noted earlier in this section, a range of aviation-related businesses with a high propensity to ship by air could concentrate around airports or along airport corridors to take advantage of access to the airport facilities [28] in the manner of the normative models of airport-led development. However, it should be reiterated that this pattern might not necessarily exist in all contexts.

Several empirical studies directly or indirectly analyzed the airfreight catchment of airports, which can span different geographical scales mentioned above [8] and also depicted in Figure 2. Some researchers assert that the airport airfreight catchment is much broader than the passengers' catchment because the area could be served by road feeder services (for the ground leg) that are not available in the passenger networks [29]. Large hub carriers also contend for airfreight via road-based feeder systems essential for conveying consignments (from other airports or consolidation points) to the applicable hubs [30].

In analyzing the airfreight catchment of several European airports, Boonekamp [32] found that that 95% of the freight processed at Amsterdam Schiphol International Airport originated within 1250 km of the said airport, which extended to neighboring countries. Further pointing to the existence of multi-country catchments, Heinitz et al. [33] investigated the spatial configuration of the catchment of the air cargo hubs in Europe, where it was discovered that most road feeder service connections stretched from as far as England to Italy. Complementing the findings of Boonekamp [32] on the catchment of Amsterdam Schiphol Airport, the authors identified a dense schedule between Schiphol, London Heathrow, and Paris Charles De Gaulle airports, and hub airports such as Frankfurt, Copenhagen, Milan Malpensa, and Vienna were also found to attract additional air cargo demand through surface transport. Boonekamp and Burghouwt [30] argue that large hub carriers compete for airfreight through the road feeder networks, which are essential for transporting the shipments (from other airports or consolidation areas) to the respective hubs.

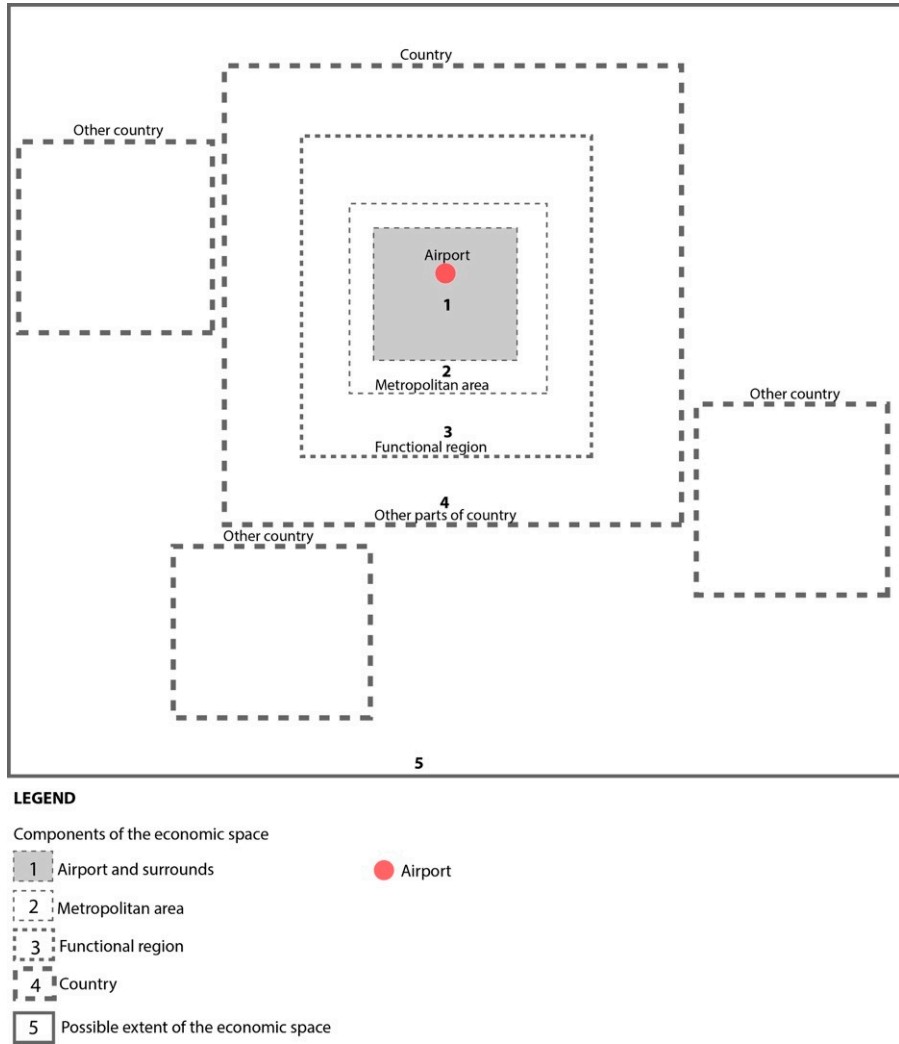

**Figure 2.** Geographical scales of airport catchment. Adapted from Mokhele [31].

Focusing on one of the airports mentioned above at a different geographical scale, Hoare [34] mapped Heathrow Airport's share of freight to approximate the extent of the catchment. It was uncovered that the breakpoint for the agents' shipments was around 240 km. Reflecting the monopoly of Heathrow Airport within the southeast region, it was only beyond the region's bounds that it was feasible for the firms to ship through other airports. In analyzing the airfreight flows relative to Hong Kong International Airport, Zhang [35] discovered that although Hong Kong was a significant source of cargo, its significance rested primarily on its role as the hub for cargo originating from the broader Pearl River Delta region. For such traffic, Hong Kong Airport faced competition from the neighboring airports, which shared the catchment area of the region. As such, Hong Kong Airport, was not a monopoly in the region, unlike Heathrow Airport in the manner of the findings of Hoare [34].

### 2.3. Factors That Influence Location-Choice of Logistics Firms

Identifying location factors for logistics facility development can provide insights for formulating appropriate logistics-related land-use policies [36]. Factors that influence the location-choice decisions of logistics facilities are diverse and include land availability and affordability, availability of transport infrastructure, level of economic development, availability of labor, and land-use planning [37]. As depicted in Figure 3, these factors can be grouped into the interrelated categories of resource endowments, economic factors, and policies and regulations [38].

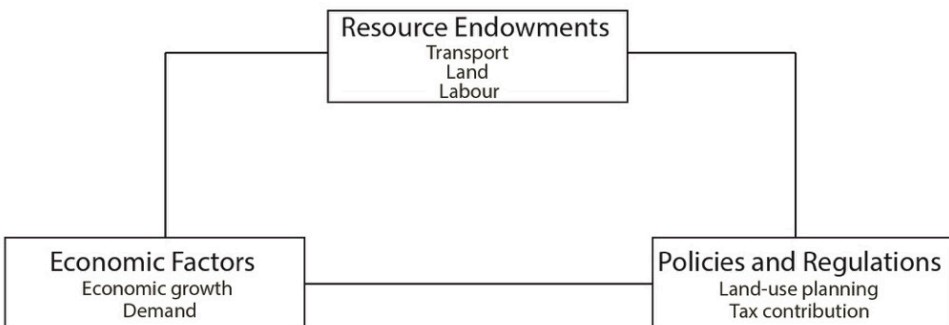

**Figure 3.** Factors influencing location-choice decisions. Adapted from Xiao et al. [38].

Regarding land availability and affordability, developable land has become a critical factor in the location choice of logistics firms. As land in the urban core is in short supply, the hotspots of logistics facility development have shifted from the central urban areas to the peripheral areas, which have ample land for development [37]. Logistics firms can only afford smaller land rents than offices, and retail projects can pay [39]. Therefore, peripheral zones near airports, for instance, attract space-intensive activities such as warehousing [40].

The literature has shown that transportation accessibility plays a vital role in influencing the location of logistics firms [38,39,41–49]. Good transport infrastructure is crucial for, among other things, expanding the market for logistics establishments and improving logistics companies' overall efficiency. Due to their strong dependence on large-scale transportation infrastructure, logistics firms tend to locate near railway stations, seaports, and airports. This shows that location choices increasingly favor zones with better transportation availability [35,43].

The spatial distribution of potential consumers or clients also influences the choice of location for logistics firms. In order to provide logistics services in a timely and efficient manner, logistics facilities are often positioned geographically close to their customers, which could include manufacturing firms [37]. Historically, logistics facilities were associated with manufacturing because the demand came mainly from manufacturing firms that needed to store their inputs and products [50,51].

Furthermore, an area's overall economic development and attractiveness have been found to affect logistics firms' location-choice decisions [37].

Another essential factor that should be considered in the logistics facility location choice is the availability and cost of the labor force. Because logistics establishments typically require many workers to maintain operations, the availability of the labor force in the surrounding areas could be a crucial factor in the location-choice decision-making processes [37].

Development policies and plans formulated by the government are also fundamental factors in the location-choice decision making of logistics firms. For instance, governments can encourage the establishment of logistics clusters in specific areas by increasing the supply of land devoted to logistics facilities or by formulating preferential development policies for logistics facilities [37]. This discussion shows that the spatial distribution of logistics establishments is not only subject to the location-choice behaviors of firms but could also be influenced by land-use planning policies and guidelines [10].

### 2.4. Analysis of the Literature

The literature overviewed above provides insights into the interrelated aspects of the airfreight catchment of airports [25–35], the relationship between airports and the surrounding metropolitan area or region [3,14,19–24], and factors that influence the location-choice decisions of logistics firms [36–49]. Notably, the latter two themes of the literature could directly or indirectly influence the geographic scope of the airfreight catchment of airports. However, although the three strands of the literature mentioned above are insightful in their own right, the interconnections are not explicitly explored by the existing

scholarship. It can, therefore, be argued that extensions are required to the knowledge of the determination of the airfreight catchment of airports amid the diverse factors that influence location-choice decisions.

Further empirical research is thus required on, among other things, the spatial economic attributes of logistics facilities, which are not only positioned near airports. The normative and empirical descriptive literature overviewed in this section primarily identified the radii (understood to represent the airport's catchment) without analyzing the logistics firms' spatial economic attributes or mapping the locational patterns of airfreight-related logistics companies within the identified catchment area.

The paper, therefore, intends to explore the nuances between the airport's relationship with the surrounding territory and the factors that influence location-choice decisions by identifying and characterizing airfreight-related firms and, accordingly, determining the airfreight catchment of airports. The paper specifically focuses on a metropolitan or municipal scale, hoping that future research on the airfreight catchment of airports in the global south will improve and extend the analyses to other geographical scales, intra-country, and inter-country.

### 3. Study Area and Methods

#### 3.1. Study Area and Data Sources

The paper focuses on the study area of the City of Cape Town municipality in the Western Cape province, South Africa (Figure 4). The study area accommodates Cape Town International Airport (CTIA), the second-busiest airport in South Africa in terms of the number of passengers and the volume of cargo handled [4]. Because CTIA is positioned in a geographically isolated location at the bottom of South Africa and the African continent, it operates as a terminal [52], unlike other major international airports that function as hubs. It can be argued that the isolated position of the City of Cape Town impacts logistics costs, which are high in the Western Cape province at approximately 16% of the gross domestic product [53]. Utilizing air transport could reduce logistics costs and the externalities of moving freight. The isolated location of the City of Cape Town makes it an apt study area for characterizing airfreight-related logistics firms.

The discussion of logistics in the City of Cape Town cannot be complete without mentioning the country's second-busiest seaport, the Port of Cape Town, which is approximately 20 km west of CTIA. Although airports and seaports have similar functions in terms of the interchange between different modes of transport, cargoes transported by air include time-sensitive low-weight and high-value goods [13].

As stated in the Introduction, the study analyzed factors that influenced the placement of logistics firms in different areas of the municipality, the extent to which logistics firms utilized CTIA for airfreight purposes, and the association between airfreight-related logistics firms and the general attributes of logistics firms in the municipality.

Because logistics firms typically locate in industrial zones, the study focused on the firms situated in the municipality's primary industrial nodes. As highlighted in the Literature Review section, logistics facilities have traditionally been associated with manufacturing businesses because logistics demand came primarily from firms that needed to store their inputs and products [50]. Although it is not an industrial area, it was crucial to include Cape Town's central business district (CBD) in the study because it is the economic core of the City of Cape Town municipality.

The data on the logistics firms within the City of Cape Town were obtained from AfriGIS in the geo-referenced geographic information system (GIS) shapefile format. Following the data cleaning process (removing the duplicate entries and omitting non-logistics entries), the dataset remained with 937 entries for logistics firms. The entry for each firm included, among other attributes, the firm's name, latitude and longitude details, a description of the economic activities conducted, and the physical or street address.

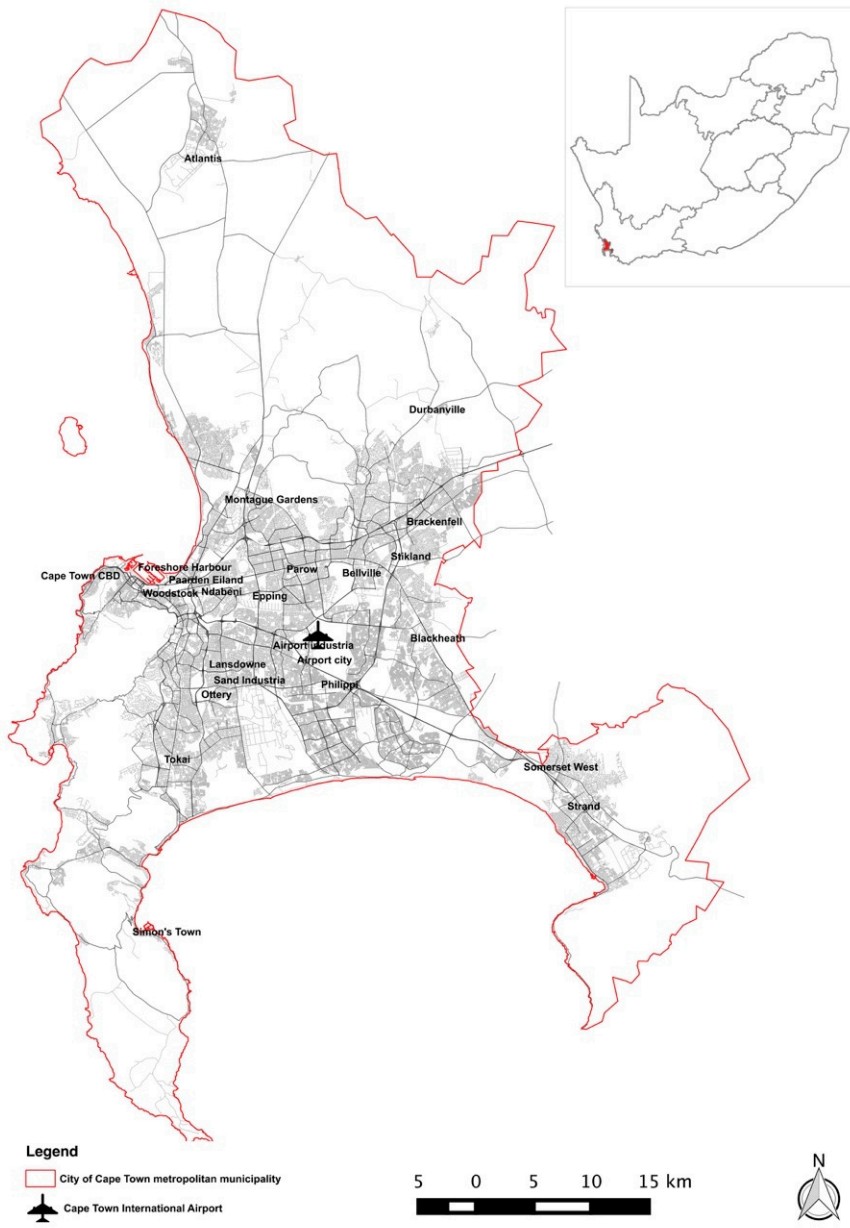

**Figure 4.** Study area.

*3.2. Descriptive Survey*

The study was based on a quantitative design, which entailed a survey and spatial analysis as outlined in the subsections below. The study adopted a descriptive survey approach, which intended to learn about the population of logistics firms in the City of Cape Town by analyzing its sample [54]. Surveys are employed in the studies with individuals as units of analysis, which in the study were the individual logistics firms situated in the primary industrial and economic nodes in the City of Cape Town municipality. Because of the lack of longitudinal data, the study utilized a cross-sectional survey approach, which involved making observations of a sample at one point in time, as opposed to longitudinal studies, which could permit the observation of the same phenomenon over an extended period [54–56]. The paper is anticipated to form the basis for future longitudinal studies on the positioning of logistics firms in the City of Cape Town.

*3.3. Sampling*

The list of 937 logistics firms obtained from AfriGIS (refer to Section 3.1) served as a sampling frame for the study. The sampling was conducted using Stata version 15 [57],

where stratified random sampling was considered appropriate because it would ensure that the firms in different industrial and economic nodes had an equal chance of being sampled, thus reducing sampling bias [58]. Different industrial and economic nodes in the municipality were used as strata, wherein a fixed number of seven logistics firms were randomly selected in each stratum. In cases where the number of logistics firms in an industrial or economic node was below seven, all logistics firms were sampled. A total of 110 firms were sampled, and 66 participated in the survey, equating to a response rate of 60%.

### 3.4. Questionnaire Development and Data Collection

The survey data were collected through telephonic interviews in October and November 2021 using a structured questionnaire containing a range of open-ended and closed-ended questions. As the study was conducted during the era of restrictions related to the COVID-19 pandemic, telephonic interviews were considered appropriate instead of face-to-face interviews. The questionnaire entailed two main sections: the first was on the logistics firms' general characteristics and locational behavior, and the second was on the airfreight-related attributes of the logistics firms (Table 1). The information required from the firms' representatives included, among others, the volume of cargo that was shipped through CTIA, the destination of the cargo, the frequency of shipping, the size of the firm (in terms of the number of employees), factors that influenced location-choice decisions, and the year of establishment of the firms at the premises occupied during the study. Additional questions were included on the impact of the COVID-19 pandemic on logistics firms.

**Table 1.** Main information on the questionnaire.

| Data Categories | Specific Data Required |
| --- | --- |
| Locational behavior and general characteristics | Description of the firms' logistics activities |
| | Ownership of the premises |
| | Year of establishment at the current premises |
| | Reasons for relocating from previous premises |
| | Location-choice reasons |
| | Number of employees on the premises |
| Airfreight-related characteristics | Shipping/receiving cargo through Cape Town International Airport |
| | Percentage of goods transported via the airport |
| | Frequency of shipping/receiving goods through the airport |
| | The volume of cargo shipped/received |
| Impact of COVID-19 | Impact of COVID-19 on the operations of logistics firms |

The names and street addresses of the sampled firms were used to search the telephone numbers of the firms on the Internet. Telephone calls were then made to the firms on the sample list, the aim of the study was presented, and enquiries were made regarding the relevant person who could be interviewed within the firm. Each interview lasted between five and fifteen minutes, during which the interviewers captured the responses on a printed copy of the questionnaire. The interview was deliberately designed to be brief, as the socio-economic climate of the distress caused by the COVID-19 pandemic was not conducive to extended interviews.

### 3.5. Data Analysis

The data collected through the survey interviews were transferred from the hard copy questionnaires to Microsoft Excel, coded, and loaded onto IBM's statistical package for social sciences (SPSS) version 28 [57] and Stata version 15 [59]. For open-ended questions, the responses were grouped into themes informed by the patterns identified from the

responses. The descriptive statistics, such as frequency distributions and cross-tabulations, were performed using Stata.

Further inferential statistical analysis was undertaken in Stata to examine the association between airfreight-related logistics firms and the general attributes of logistics firms in the City of Cape Town. As the realized sample for this study was small (n = 66), Fischer's exact test was considered more appropriate than chi-square to analyze the association between variables. A *p*-value of 0.05 or less was used for statistical significance.

### 3.6. Spatial Analysis

The GIS data from AfriGIS formed the basis of spatial analysis conducted in ArcGIS 10.8 [60] and QGIS 3.16 [61]. This analysis was undertaken to map the airfreight-related logistics firms' locational patterns relative to the airport. The survey data were incorporated into the spatial analysis through the longitude and latitude information contained in the underlying spatial dataset of the logistics firms (Section 3.1).

## 4. Results

This section discusses the results of the statistical and spatial analyses conducted. The discussion is organized around the three research objectives presented in the Introduction. The objectives addressed factors that influenced the positioning of logistics firms in the City of Cape Town municipality, the linkages of logistics firms with Cape Town International Airport, and the association between airfreight-related logistics firms and the general attributes of logistics firms in the municipality.

### 4.1. General Characteristics and Factors That Influence the Location of Logistics Firms

As reflected by the responses in Table 2, logistics firms were involved in multiple economic activities. For the first category of responses, distribution or transportation topped the list with 20 responses, followed by warehousing and manufacturing with 15 and 10, respectively. The supplier of goods recorded nine responses, while the courier recorded the least with one response. For the second category, distribution or transportation dominated with eight, followed by manufacturing and suppliers of goods, each accounting for five. The findings in the first two categories reflect the close connection between logistics facilities and manufacturing acknowledged in the literature [50,51]. Showing the magnitude of this connection, it should be noted that the close-ended question on the questionnaire (see Section 3.4) did not include the option 'manufacturing'. However, manufacturing had to be separated from the 'other' category because many respondents mentioned that their firms were involved in manufacturing activities. Packaging dominated the third category with six responses, followed by courier with two responses. For the fourth category, couriers dominated with five, while suppliers of goods recorded one. Concerning the last category, suppliers of goods recorded five, while other activities did not have a record.

**Table 2.** Multiple responses to the activities of logistics firms.

| Activity | Category 1 | Category 2 | Category 3 | Category 4 | Category 5 |
|---|---|---|---|---|---|
| Distribution/transportation | 20 | 8 | | | |
| Warehousing | 15 | | | | |
| Manufacturing | 10 | 5 | | | |
| Supplier of goods | 9 | 5 | | 1 | 5 |
| Other | 6 | | | | |
| Packaging | 5 | 3 | 6 | | |
| Courier | 1 | | 2 | 5 | |
| Total | 66 | 21 | 8 | 6 | 5 |

The size of logistics firms was analyzed using the number of employees as a proxy for size [42,62]. Table 3 shows that the majority (38.9%) of the firms had between 10 and 49 employees, which are regarded as small firms according to the guidelines set by the

Republic of South Africa [63]. Very small firms, those with between 5 and 9 employees, followed with 18.5%. The firms with between 50 and 99 employees (medium firms) and those with 100 or more employees (large firms) had 14.8% each. Micro firms, those with between 1 and 4 employees, represented the least proportion with 13.0%. The findings show that, despite the diverse activities of the establishments, most logistics firms were small.

**Table 3.** The size of logistics firms based on number of employees.

| Number of Employees | n | % |
|---|---|---|
| 1–4 (Micro) | 7 | 13.0 |
| 5–9 (Very small) | 10 | 18.5 |
| 10–49 (Small) | 21 | 38.9 |
| 50–99 (Medium) | 8 | 14.8 |
| 100+ (Large) | 8 | 14.8 |
| Total | 54 | 100.0 |

Against the backdrop of the multiple activities of the logistics firms and the dominance of small firms, it was essential to ascertain the age of the firms concerning the year of establishment at the premises occupied during the study. Table 4 indicates that most (25) of the respondent logistics firms were established at their premises between 2010 and 2019. This group was followed by those established between 2000 and 2009, with 15. Six logistics firms were established between 1990 and 1999, while four were established between 1980 and 1989. The findings reflect that most logistics firms were young, as they had recently located at the premises occupied during the study.

**Table 4.** Year in which the logistics firms were established.

| Year | n | % |
|---|---|---|
| 1980–1989 | 4 | 8.0 |
| 1990–1999 | 6 | 12.0 |
| 2000–2009 | 15 | 30.0 |
| 2010–2019 | 25 | 50.0 |
| Total | 50 | 100.0 |

In light of the findings above that most logistics firms had located at their current premises relatively recently, it was essential to analyze the locational behavior of the firms. Regarding whether the logistics firms had always been located at the premises occupied during the study, 41.1% indicated so. In comparison, 58.9% reported that they had not always been located at their current premises (Figure 5). This shows that although most logistics firms had recently located at the premises occupied during the study, they were not necessarily young overall, as some had relocated from elsewhere.

To ascertain the reasons behind the choice of location by the logistics firms, the respondents were asked two open-ended questions: (1) the reasons the company moved from the previous location for those who had not always been at their current premises; and (2) the reasons the company located at their current premises. Figure 6 shows that the majority (10) of the logistics firms moved due to the previous premises being small and no longer meeting the requirements of the firms, followed by those who bought property or constructed their premises, with four. The rental of the previous premises being expensive and relocation due to COVID-19 accounted for one each.

In addition to the 'push' factors presented above, it was equally essential to analyze the 'pull' factors that attracted the logistics firms. Reflecting the diversity of factors that influenced location-choice decisions of logistics firms [37] and aligned with the reason for moving due to the insufficient size of the premises, Figure 7 shows that the majority (9) of the logistics firms located in their current premises because of the availability of bigger premises. Showing the role of accessibility in the location choice of logistics firms [37], the

centrality or accessibility of the logistics firms' premises was second with five responses. Relatedly, one logistics firm was located at its current location because of its proximity to the harbor. Notably, in response to an open-ended question, none of the firms explicitly mentioned CTIA as a factor that influenced location-choice decisions.

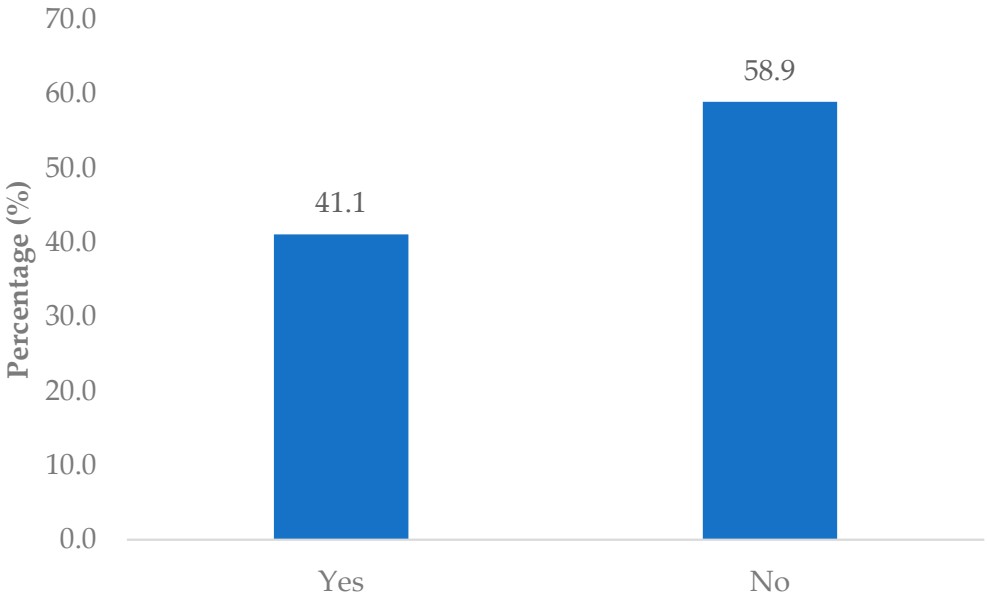

**Figure 5.** The logistics firm has always been located at the current premises (n = 56).

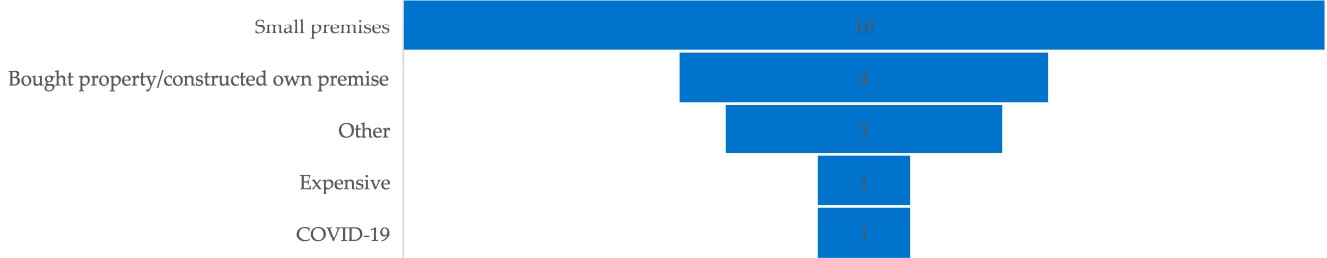

**Figure 6.** Reasons the logistics firm moved from the previous location (n = 19).

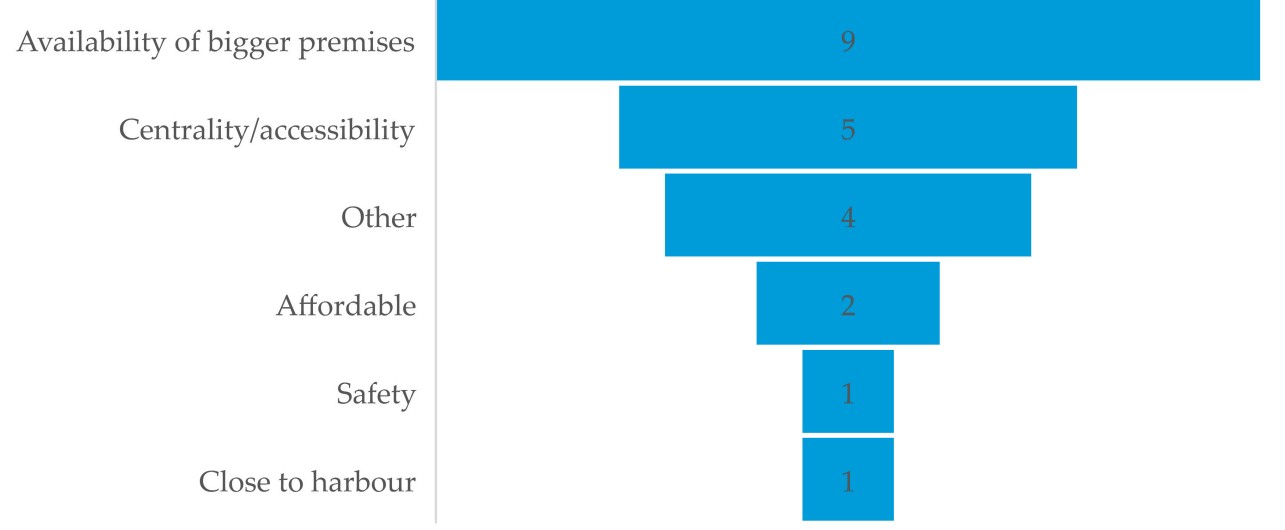

**Figure 7.** Reasons for the logistics firm's location at the current location (n = 22).

Given the importance of property-related considerations in the locational behavior of logistics firms, it was essential to analyze the ownership of the premises occupied during the study. This analysis could provide insight into the footlooseness of the logistics firms and, in part, ascertain the ability of the current premises to retain the firms. The literature notes that outsourcing logistics functions, among other reasons, has changed the property ownership structure from owning to leasing warehousing facilities, implying that logistics firms could respond quickly to internal and external demands to relocate or build new facilities [49]. Figure 8 reflects that 39.7% of the logistics firms owned the premises they occupied, while the majority (60.3%) indicated that they did not own the premises. The findings show that the locational patterns of logistics firms mapped in the study could easily change because of possible relocations due to changing internal and external demands.

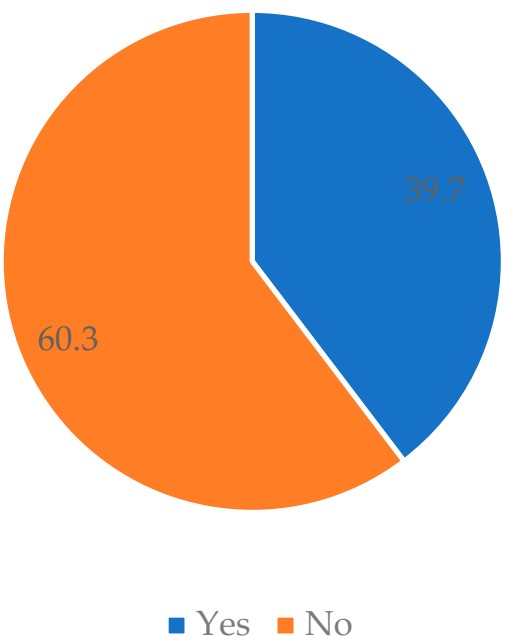

■ Yes  ■ No

**Figure 8.** Logistics firm owned the premises it occupied (n = 58).

*4.2. Linkages of Logistics Firms with Cape Town International Airport*

Despite the airport not being explicitly mentioned as a location-choice factor, it was essential to establish the linkages of the logistics firms with CTIA to address the second research objective and ascertain the airport's catchment at a municipal scale. Sixteen logistics firms reported shipping or receiving cargo through CTIA, which translates to a quarter (25.0%) of the firms. The majority (75.0%) of the logistics firms in the City of Cape Town reported not shipping or receiving cargo through CTIA (Figure 9). The findings show that logistics firms in the municipality predominantly used other modes of transport. The presence of the country's second-busiest port, the Port of Cape Town, is noted in this regard (see Section 3.1).

Figure 10 shows that the potential airfreight catchment of CTIA (at a municipal scale) extends to about a 20 km radius of the airport. Compared to the literature's findings, the geographical extent of the potential metropolitan catchment of CTIA is small. Most firms that reported directly using the airfreight services were situated beyond a 10 km radius of the airport, with only two airfreight-related firms out of the 16 positioned within a 5 km radius. The results do not support the literature's assertion that aviation-related businesses increasingly concentrate near airports. However, it is acknowledged that they may be positioned along transport routes that provide access to the airport [28]. The findings also do not depict the logic of a radial network of logistics facilities around the airport [14,22]. Although there was no clear spatial pattern of the location of logistics firms concerning size, the two large logistics firms were located within a 10 km radius of CTIA (Figure 10).

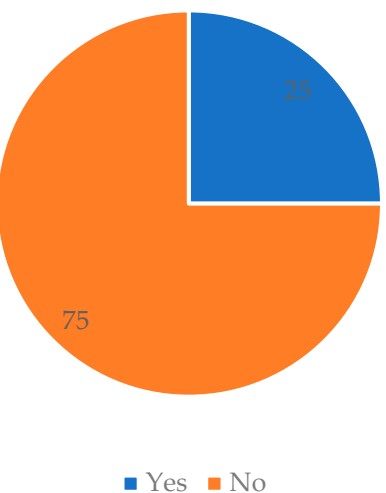

■ Yes ■ No

**Figure 9.** The logistics firm ships or receives cargo through CTIA (n = 64).

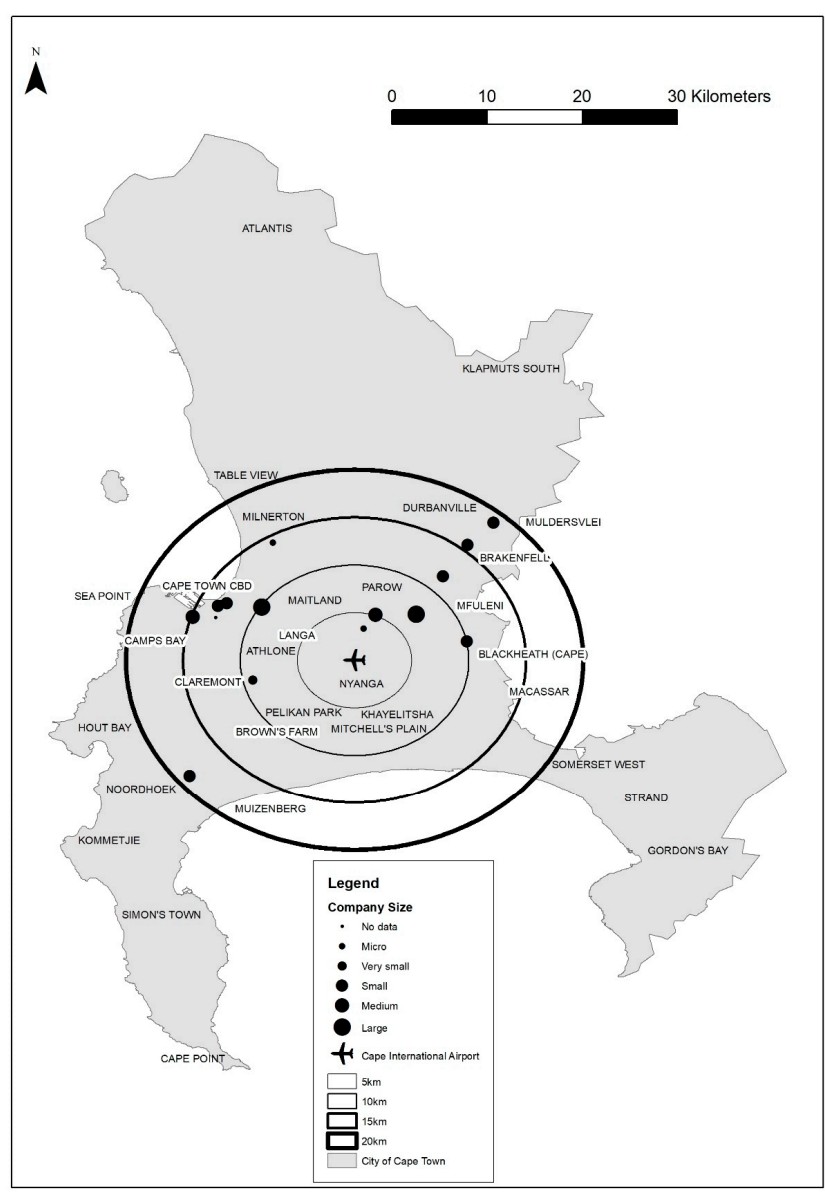

**Figure 10.** Logistics firms that utilized CTIA and their size (n = 16).

Of the 16 logistics firms that reported using CTIA, the majority (7) were involved in distribution or transportation activities, followed by those involved in warehousing, with four (Figure 11). Manufacturing and suppliers of goods accounted for two logistics firms each. It is noted that the firms involved in courier activities did not report that they utilized the airport for airfreight purposes. This shows that such firms focused on parcel transfers within the municipality and region and were not responsible for the logistics of shipping parcels through the airport.

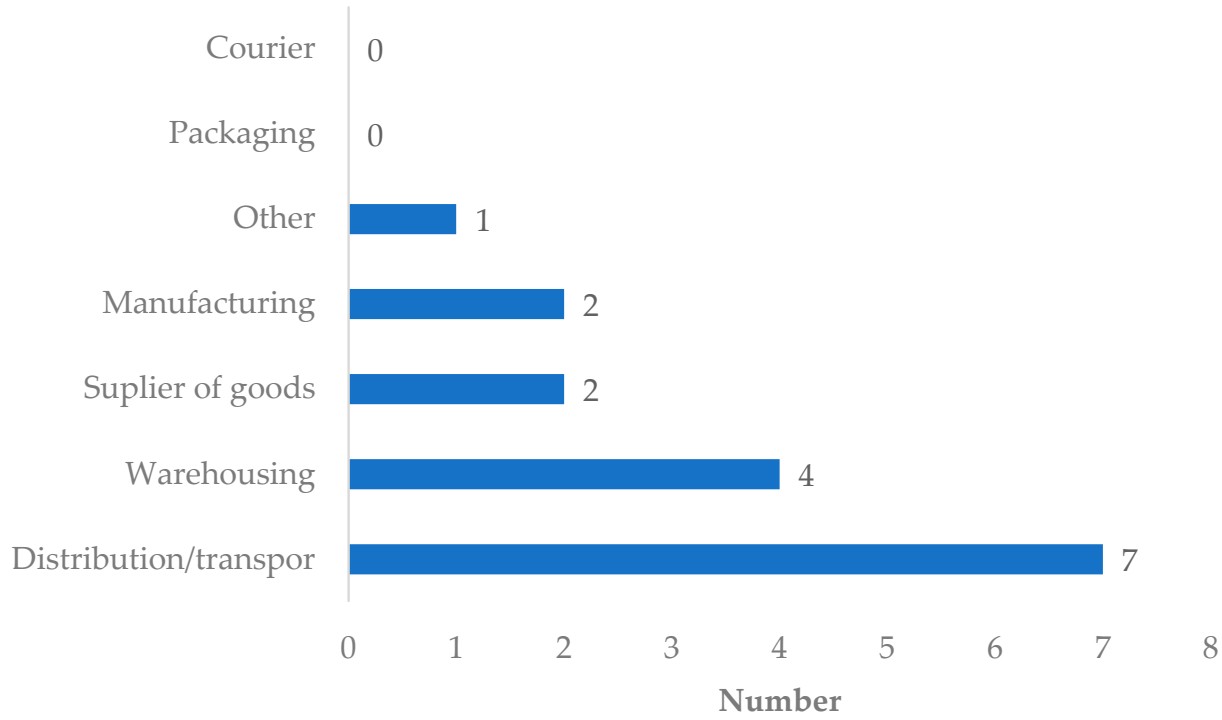

**Figure 11.** Activities of the logistics firms that ship or receive cargo through CTIA (n = 16).

It was essential to go beyond merely identifying firms that used the airport and analyze the extent of its utilization. Six airfreight-related logistics firms reported shipping or receiving cargo through CTIA monthly (Figure 12). These were followed by four firms that indicated they used the airport daily, and those that indicated weekly or fortnightly use of the airport recorded three and one, respectively. Reflecting the airport's significance for the operations of logistics firms, the results show that airfreight-related logistics firms used airport cargo services frequently. Although, for a comprehensive dissection of the airport-relatedness of logistics firms, the survey questionnaire had a question on the quantity of cargo shipped through the airport, the respondent firms could not accurately provide that information; hence it was not included in the analysis.

### 4.3. COVID-19 Effects on Logistics Firms' Operations

As the survey was conducted during COVID-19 lockdown restrictions, the respondents were asked whether their firms' operations were affected by COVID-19. Figure 13 indicates that the operations of three-quarters (75.0%) of the logistics firms were affected by COVID-19, and a quarter (25.0%) reported that their operations were unaffected by the pandemic. Although this study did not obtain further details of the impact of COVID-19, it has been reported elsewhere that factors that affected logistics firms during COVID-19 included labor shortage, a shortage of transportation capacity, a lack of safety, a disruption of the logistics network, a sharp drop in logistics demand, a change of service mode, disgruntled customers, and an increase in operating costs [64,65]. The findings that a quarter of the firms were unaffected by COVID-19 can, to some extent, be supported by Atayah et al. [66],

who found that some logistics firms performed well during the COVID-19 pandemic period compared to the prior 10 years.

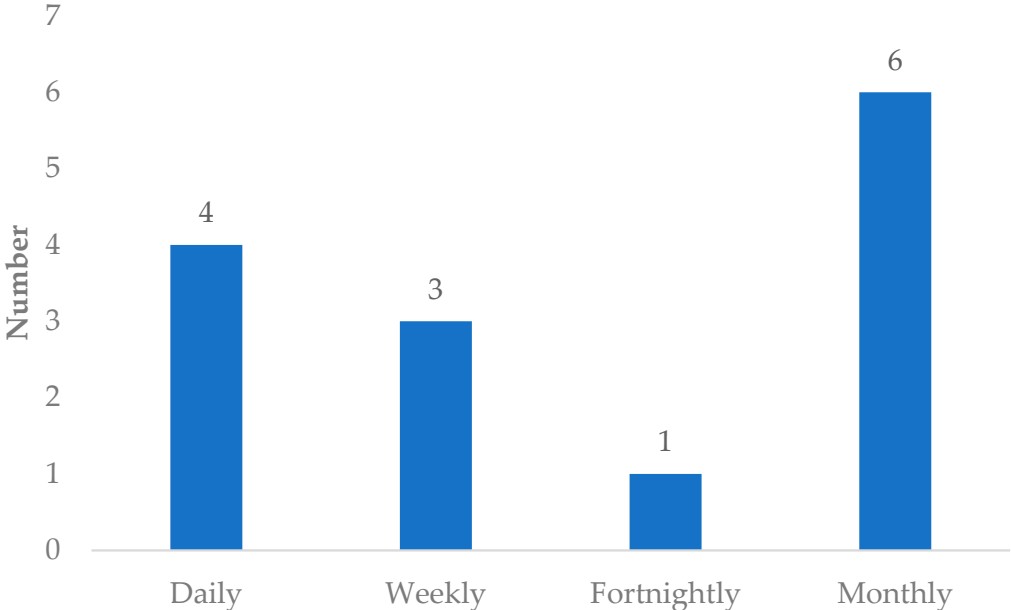

**Figure 12.** The logistics firm's frequency of shipping or receiving cargo through CTIA (n = 14).

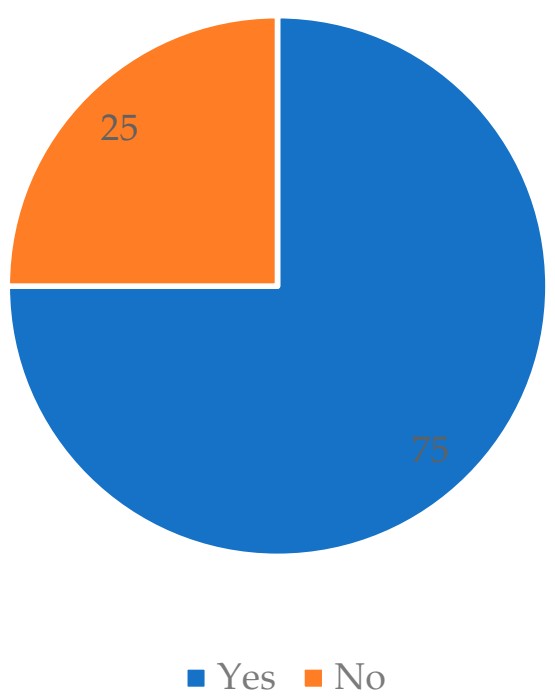

■ Yes  ■ No

**Figure 13.** The logistics firm's operations affected by COVID-19 (n = 60).

*4.4. Association between Airfreight-Related Logistics Firms and the General Attributes of Logistics Firms in the City of Cape Town*

Table 5 shows that the Fisher's Exact Test *p*-value was 1.000, which was insignificant, $p > 0.05$. Therefore, there was no significant association between whether a logistics firm owned the premises it occupied and whether it shipped or received cargo through CTIA.

**Table 5.** Fisher's exact results for the association between whether the company owned the premises it occupied and whether it shipped or received cargo through CTIA.

| Does Your Company Own the Premises Occupied at This Location? | Does Your Company Ship/Receive Cargo through CTIA? | | Total |
|---|---|---|---|
| | **Yes** | **No** | **Total** |
| Yes | 6 | 16 | 22 |
| No | 9 | 25 | 34 |
| Total | 15 | 41 | 56 |

Fisher's exact = 1.000, 1-sided Fisher's exact = 0.592.

Table 6 shows that the Fisher's exact test *p*-value was 1.000, which was insignificant, *p* > 0.05. Therefore, no significant association existed between when a logistics firm was established and whether it shipped or received cargo through CTIA.

**Table 6.** Fisher's exact results for the association between the year the company was established and whether it shipped or received cargo through CTIA.

| Year in Which Company Was Established | Does Your Company Ship/Receive Cargo through CTIA? | | Total |
|---|---|---|---|
| | **Yes** | **No** | **Total** |
| 1980–2009 | 7 | 18 | 25 |
| 2010–2019 | 8 | 17 | 25 |
| Total | 15 | 35 | 50 |

Fisher's exact = 1.000, 1-sided Fisher's exact = 0.500.

Table 7 shows that the Fisher's exact test *p*-value was 0.231, which was insignificant, *p* > 0.05. Therefore, there was no significant association between whether a logistics firm had always been located at the current premises and whether it shipped or received cargo through CTIA.

**Table 7.** Fisher's exact results for the association between whether the company had always been located at the current premises and whether it shipped or received cargo through CTIA.

| Has Your Company Always Been Located at the Current Premises | Does Your Company Ship/Receive Cargo through CTIA? | | Total |
|---|---|---|---|
| | **Yes** | **No** | **Total** |
| Yes | 4 | 19 | 23 |
| No | 11 | 22 | 33 |
| Total | 15 | 41 | 56 |

Fisher's exact = 0.231, 1-sided Fisher's exact = 0.154.

Table 8 shows that the Fisher's exact test *p*-value was 0.748, which was insignificant, *p* > 0.05. Therefore, there was no significant association between the company's size and whether it shipped or received cargo through CTIA.

Table 9 shows that the Fisher's exact test *p*-value was 1.000, which was insignificant, *p* > 0.05. Therefore, there was no significant association between whether a logistics firm was affected by COVID-19 and whether it shipped or received cargo through CTIA.

**Table 8.** Fisher's exact results for the association between the size of the logistics firm and whether it shipped or received cargo through CTIA.

| How Many People Are Employed by Your Company, at Your Location? | Does Your Company Ship/Receive Cargo through CTIA? | | Total |
| --- | --- | --- | --- |
| | **Yes** | **No** | |
| 1–9 | 4 | 13 | 45 |
| 10+ | 11 | 25 | 14 |
| Total | 15 | 38 | 53 |

Fisher's exact = 0.748, 1-sided Fisher's exact = 0.426.

**Table 9.** Fisher's exact results for the association between whether the company was affected by COVID-19 and whether it shipped or received cargo through CTIA.

| Did COVID-19 Pandemic Affect the Operations of Your Company? | Does Your Company Ship/Receive Cargo through CTIA? | | Total |
| --- | --- | --- | --- |
| | **Yes** | **No** | |
| Yes | 12 | 33 | 45 |
| No | 3 | 11 | 14 |
| Total | 15 | 44 | 59 |

Fisher's exact = 1.000, 1-sided Fisher's exact = 0.496.

## 5. Conclusions

Focusing on the City of Cape Town study area, the paper aimed to characterize airfreight-related logistics firms relative to non-airfreight-related logistics firms in order to establish the airfreight catchment of Cape Town International Airport. This aim was achieved by analyzing factors that influenced the location of logistics firms in the municipality, the linkages of logistics firms with Cape Town International Airport, and the association between airfreight-related logistics firms and the general attributes of logistics firms in the City of Cape Town. Regarding the factors that influenced the placement of logistics firms, the findings revealed that the positioning of logistics firms in the municipality was influenced mainly by property-related considerations (specifically the size of the firm's premises) and centrality or accessibility considerations. The airport was not explicitly mentioned as a factor that directly influenced the positioning of logistics firms. Concerning the relationship between logistics firms and the airport, though the number of logistics firms that confirmed utilizing the airport for airfreight purposes was relatively small, it was found that airfreight-related logistics firms did not necessarily locate near the airport but were positioned within a 20 km Euclidean radius of the airport. A non-significant association was uncovered between airfreight-related logistics firms and the general attributes of logistics firms in the study area, including ownership of the premises, year of establishment, locational behavior, and the size of firms.

In formulating the spatial plans or spatial development frameworks (with a clear logistics-related strategy), the planning authorities and other stakeholders must take cognizance of the possible extent of the catchment, wherein airfreight-related logistics firms do not necessarily locate near Cape Town International Airport. Partly explaining this locational pattern, the stakeholders need to note that the spatial economic attributes of airfreight-related firms are not significantly different from those of non-airfreight-related logistics firms, implying that airfreight-related firms could locate in diverse areas of the municipality. Therefore, instead of assuming that airfreight-related firms have to be only positioned in the vicinity of the airport, the focus should be on, for instance, augmenting the airfreight flows in the municipality and accordingly improving the transport infrastructure networks to and from the airport, while noting that the locational patterns mapped

could significantly change in the future due to the demands internal and external to the logistics firms.

The main shortcoming of the study is that the analysis was limited to the City of Cape Town metropolitan scale and did not analyze linkages that transcend administrative boundaries. In addition, the realized response rate was very low; hence limited advanced statistical analysis, such as Fischer's exact test, could be explored. To further unpack the airfreight catchment of Cape Town International Airport, it is recommended that future research extend the analysis presented herein to include the broader functional region the City of Cape Town municipality is part of.

**Author Contributions:** Conceptualization, M.M. and T.M.; methodology, M.M. and T.M.; software, M.M. and T.M.; formal analysis, M.M. and T.M.; investigation, M.M. and T.M.; writing—original draft preparation, M.M.; writing—review and editing, M.M. and T.M.; visualization, M.M. and T.M.; project administration, M.M.; funding acquisition, M.M and T.M. All authors have read and agreed to the published version of the manuscript.

**Funding:** This research was funded by the National Research Foundation, grant number 120662, and the APC was funded by Cape Peninsula University of Technology.

**Institutional Review Board Statement:** The study was conducted in accordance with the Declaration of Helsinki and approved by the Research Ethics Committee of the Faculty of Informatics and Design, Cape Peninsula University of Technology (14 October 2019).

**Data Availability Statement:** Data sharing is not applicable.

**Acknowledgments:** The authors thank the respondents who took time to take part in the study. The research assistants who conducted the telephonic interviews are also acknowledged. Finally, the authors acknowledge the constructive comments of the anonymous reviewers and the editorial team.

**Conflicts of Interest:** The authors declare no conflict of interest. The funders had no role in the design of the study; in the collection, analyses, or interpretation of data; in the writing of the manuscript; or in the decision to publish the results.

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
