# Peer review of "Characterization of Airfreight-Related Logistics Firms in the City of Cape Town, South Africa"

_logistics, 2023_

Round 1

Reviewer 1 Report

Please refer to the attached pdf for comments. You may use Microsoft Edge.

One comment of particular importance is the average time taken to complete the questionnaire by a respondent. Please kindly check your data.

Thank you.

Author Response

One comment of particular importance is the average time taken to complete the questionnaire by a respondent. Please kindly check your data.

This has been amended. The time taken to complete the questionnaire was between 5 and 15 minutes.

Kindly consider the following (Comments were extracted from the .pdf):

1. A figure illustrating concentric zones could be helpful.

Figure 2 has been added.

2. Do kindly consider a figure for this section to summarize. Cite as well.

Figure 3, showing factors, has been added.

3. Since

This word has been replaced as suggested.

4. 5 minutes is unlikely sufficient to answer all the questions. Please kindly check your data.

The sentence has been revised accordingly. The time taken to complete the questionnaire was between 5 and 15 minutes.

5. Kindly state the specific variables.

The sentence has been revised. It now reads, “Further inferential statistical analysis was undertaken in Stata to examine the association between airfreight-related logistics firms and the general attributes of logistics firms in the City of Cape Town.”

6. Small letter p in italic.

Capital P has been replaced with a small p in italic.

Reviewer 2 Report

This paper determined the “Characterization of airfreight-related logistics firms in the city of cape town, south Africa”. This study aims to three research objectives, including an analysis of factors that affect the placement of logistics firms within the municipality, an analysis of linkages of the logistics firms with Cape Town International Airport, and a comparison of the airfreight-related firms with non-airfreight logistics firms. The results indicated that a quarter of the respondent logistics firms used CTIA for airfreight purposes; the potential airfreight catchment of CTIA extended to about a 20km radius of the airport. The contributions can be applied to the airfreight catchment of airport developments. It is relevant and valuable to the readers of Logistics. However, several places need to be revised. Therefore, I recommend the publication of this paper subject after major revision. The following changes to improve this paper was suggested.

  1. Introduction, the authors select Cape Town International Airport to be its background. However, I do not find any empirical problem between Cape Town International Airport and its surrounding area. Is this a particular problem? What is different between CTIA and the other international airports in the world? 
  2. Both the airport and seaport are the doors for a country. Are there any differences in logistics between the airport and the seaport logistics? What are the specific airport logistics activities? Please common them in the Literature Review.
  3. The authors should provide more information as to why they executed the sampling strategy. Why did you select Cape Town International Airport as the sampling? Did an organization sponsor the survey? Was it delivered on paper or electronically? Are there any experts who participate in this topic? Also, how about sample selection bias? Did the authors attempt to account for this effect here?
  4. The conclusions drawn in this paper are too simple. They are just some results but have yet to be excavated more deeply. Relevant conclusions are difficult to apply to the issue. Please show how we apply your result in practice. In addition, please provide more suggestions for future work.

Non

Author Response

1. Introduction, the authors select Cape Town International Airport to be its background. However, I do not find any empirical problem between Cape Town International Airport and its surrounding area. Is this a particular problem? What is different between CTIA and the other international airports in the world? 

The main difference between Cape Town International Airport and other major hub airports is that it operates as a terminal because of being positioned in a geographically isolated position at the bottom of the country and the African continent. This isolated location can negatively impact logistics costs, which are high in the Western Cape and South Africa. This discussion has been added to Section 3.1: Study Area and Data Sources.

2. Both the airport and seaport are the doors for a country. Are there any differences in logistics between the airport and the seaport logistics? What are the specific airport logistics activities? Please common them in the Literature Review.

In Section 3.1, there is now a mention of the Port of Cape Town and a general indication of the type of goods that are transported by air, i.e. time-sensitive, low-weight and high-value goods.

3. The authors should provide more information as to why they executed the sampling strategy. Why did you select Cape Town International Airport as the sampling? Did an organization sponsor the survey? Was it delivered on paper or electronically? Are there any experts who participate in this topic? Also, how about sample selection bias? Did the authors attempt to account for this effect here?

Cape International Airport Town was not ‘sampled’ from a list of airports. It was selected as a case study because of the unique characteristics of being geographically isolated at the bottom of the country and the continent.

The National Research Foundation funded the research. No organization with a vested interest in airports funded the research.

As mentioned in Section 3.4 (after Table 1), the interviews were telephonically conducted, and the responses were captured on hard-copy questionnaires.

As discussed in Section 3.3, stratified sampling would ensure that the firms in different strata had an equal chance of being selected. There is now a mention that stratified sampling reduces sampling bias.

4. The conclusions drawn in this paper are too simple. They are just some results but have yet to be excavated more deeply. Relevant conclusions are difficult to apply to the issue. Please show how we apply your result in practice. In addition, please provide more suggestions for future work.

The conclusion has been updated to mention that in drafting the spatial plans in the City of Cape Town, the possible extent of the airfreight catchment of CTIA should be factored in. Instead of assuming that airfreight-related logistics firms will only locate near the airports, efforts could be put into enhancing the freight flows between the various areas of the municipality and the airport.

The suggestion for future work on the airfreight catchment of Cape Town International Airport is that the analysis should be extended to include the broader functional region that the City of Cape Town municipality is part of.

Reviewer 3 Report

The issues taken up by the authors to present and describe are interesting from both a practical and scientific point of view. 

The authors' methodology and choice of objects are not objectionable, but the manuscript needs to be significantly improved to be suitable for publication.

1) in the section dealing with the literature on the subject, the discussion should be supplemented with citations and references to the literature on the subject, so that it is clear that the authors not only assume that literature exists, not only write that there are no practical implications and that there is a need for empirical research, but can substantiate these theses with adequate literature sources

2) the research analyses carried out are at a rather basic, weak level, lacking more involvement and presentation of more advanced statistical methods. Application to a professional journal requires their use in order to show the different facets of the research carried out

3) there is also a lack of well-developed limitations and managerial implications, a lack of in-depth discussion and justification - or rather, a lack of answers to the research questions or theses raised. 

Moderate editing of English language

Author Response

The authors' methodology and choice of objects are not objectionable, but the manuscript needs to be significantly improved to be suitable for publication.

The introduction has been improved by presenting an overarching aim of the study, which links to the preceding discussion in the introduction. The aim is achieved by addressing the three objectives outlined in the previous version of the manuscript. The third objective has been amended to "Analysis of the association between airfreight-related logistics firms and the general attributes of logistics firms in the City of Cape Town".

1) in the section dealing with the literature on the subject, the discussion should be supplemented with citations and references to the literature on the subject, so that it is clear that the authors not only assume that literature exists, not only write that there are no practical implications and that there is a need for empirical research, but can substantiate these theses with adequate literature sources

Citations have been added to Section 2.4, Analysis of the Literature. This is the literature presented in Sections 2.1, 2.2 and 2.3. Additional literature has also been included. 

2) the research analyses carried out are at a rather basic, weak level, lacking more involvement and presentation of more advanced statistical methods. Application to a professional journal requires their use in order to show the different facets of the research carried out

Although analysis taken in this paper is more descriptive in nature, some advanced analyses, which tested the significance of association, were performed using Fischer’s exact test, which is suitable for small sample sizes. The small sample size did not allow further advanced statistical analysis other than this. This limitation has been added to the limitations or shortcomings of this study in the conclusion.

3) there is also a lack of well-developed limitations and managerial implications, a lack of in-depth discussion and justification - or rather, a lack of answers to the research questions or theses raised. 

The third objective has been revised to "Analysis of the association between airfreight-related logistics firms and the overall characteristics of logistics firms".

Answers to the overarching research aim and the objectives have now been summarised in the first paragraph of the conclusion.

Round 2

Reviewer 2 Report

This paper determined the “Characterization of airfreight-related logistics firms in the city of cape town, south Africa”. This study aims to three research objectives, including an analysis of factors that affect the placement of logistics firms within the municipality, an analysis of linkages of the logistics firms with Cape Town International Airport, and a comparison of the airfreight-related firms with non-airfreight logistics firms. The results indicated that a quarter of the respondent logistics firms used CTIA for airfreight purposes; the potential airfreight catchment of CTIA extended to about a 20km radius of the airport. The contributions can be applied to the airfreight catchment of airport developments. It is relevant and valuable to the readers of Logistics.

The authors have modified and revised all the requests. I have no further questions. Thank you.and revised all the requests. I have no further questions. Thank you.

Non.

Reviewer 3 Report

Accepted

no comments